# End-to-End Learning and Intervention in Games

**Jiayang Li**
Northwestern University
jiayangli2024@u.northwestern.edu

**Jing Yu**
Northwestern University
jingyu2021@u.northwestern.edu

**Yu (Marco) Nie**
Northwestern University
y-nie@northwestern.edu

**Zhaoran Wang**
Northwestern University
zhaoranwang@gmail.com

## Abstract

In a social system, the self-interest of agents can be detrimental to the collective good, sometimes leading to social dilemmas. To resolve such a conflict, a central designer may *intervene* by either redesigning the system or incentivizing the agents to change their behaviors. To be effective, the designer must anticipate how the agents react to the intervention, which is dictated by their often unknown payoff functions. Therefore, *learning* about the agents is a prerequisite for intervention. In this paper, we provide a unified framework for learning and intervention in games. We cast the equilibria of games as individual layers and integrate them into an *end-to-end* optimization framework. To enable the backward propagation through the equilibria of games, we propose two approaches, respectively based on explicit and implicit differentiation. Specifically, we cast the equilibria as the solutions to *variational inequalities* (VIs). The explicit approach unrolls the projection method for solving VIs, while the implicit approach exploits the sensitivity of the solutions to VIs. At the core of both approaches is the differentiation through a projection operator. Moreover, we establish the correctness of both approaches and identify the conditions under which one approach is more desirable than the other. The analytical results are validated using several real-world problems.

## 1 Introduction

The history of human societies may be viewed as an evolutionary process through which countless self-interested individuals learn to cooperate with each other [20]. While human self-interest can be channeled towards socially desirable ends, interventions—in the form of laws, social norms and incentives—are often required. Indeed, even the "invisible hand" of Adam Smith would not work without proper regulations and policing. This process continues, as it uncovers and resolves previously unknown or non-existent conflicts between self- and collective interest. For example, the potential conflict between overpopulation and welfare states has been heatedly debated among biologists, social scientists, philosophers and alike [21, 16]. In economics, externalities (a.k.a. neighboring effects) lead to market failures because self-interested agents do not bear the cost/benefit of their actions in its entirety. Lloyd's common devastated by excessive grazing [34] and Pigou's road jammed by selfish drivers [50] are two classical examples. More recently, Braess [10] shows expanding a road network could worsen traffic congestion. This paradoxical phenomenon, related closely to the price of anarchy [28], demonstrates vividly how unregulated self-interest may be detrimental to the social good. In this paper, we develop a general framework aiming to regulate various systems comprised of self-interested agents.

*Game theory* is often used to determine the most likely outcomes of a system in which agents pursue self-interest and interact with each other [27, 18]. Although a game-theoretic model of the real-world

is a simplification, it can be useful for not only explaining and predicting system outcomes, but also engineering desired ones [43]. For example, many phenomena in ecosystems can be explained as the outcome of the population, in the game of survival, adopting an *evolutionarily stable strategy* [55]. Stackelberg games [61], which concern the strategic interactions between leaders and followers, have seen applications in economics [5], national security [51] and environment protection [65]. Congestion game [52], in which the utility of agents depends on a resource whose cost increases with the number of users, is another example. Many social and engineering systems can be modeled as a congestion game, with applications ranging from planning transportation infrastructure [6], managing wireless communication networks [30], to operating ride-hail companies [12].

In a game-theoretic system, we define the central designer as an authority whose action can influence the outcome of the game. The central designer can *intervene* in order to guide the self-interested behavior toward a socially desirable outcome. There are generally two types of interventions: redesign the system or modify the payoffs of the agents through incentives. Take transportation planning as example. To alleviate congestion, the owner of the road network (typically the "government"), has the power to add capacities at selected locations in the network [37, 64]. Alternatively, it may charge road users a "congestion toll", in the spirit of Pigou [50] and Vickrey [60], to incentivize them to change travel behaviors (route, departure time, mode, etc.). In order to intervene effectively, the central designer must anticipate the reaction of the agents, which is dictated by their often *unknown* payoff functions. Thus, an equally important task is to infer, from empirical observations, how the agents evaluate their payoffs. To this end, the random utility theory [41] is widely used to estimate behavioral parameters of agents in marketing [40], environmental studies [59] and travel forecasting [7]. Alternatively, the learning-theoretic approach is increasingly used to learn, among other things, the optimal strategy of agents [32] or unknown parameters of games [33].

**Contribution.** This paper provides a unified framework for learning and interventions in games. It is well known that the equilibria of many games can be formulated as either a complementarity problem [66] or an optimization problem [45], and both can be interpreted as a variational inequality (VI) problem [47]. Therefore, we propose to cast the equilibria of games, in the form of VI, as individual layers in an end-to-end optimization framework. Such a general representation of the game-theoretic system in an end-to-end framework poses the challenge of performing forward and backward propagation through the VI layers.

Along the above line, our contributions are as follows. (1) We present a unified optimization model for learning and interventions in games that can be solved by gradient descent methods. (2) We devise a Newton's method for forward propagation over VI layers. Unlike other Newton-type methods for solving VI, e.g. [8, 56], we directly find the solution via root-finding. (3) We propose two methods for backward propagation over VI layers based on explicit and implicit differentiation, respectively. The explicit approach unrolls the projection methods for solving VIs, and the implicit approach performs differentiation on the fixed-point formulation of VI [26]. The implicit approach is more efficient than the explicit approach but is applicable only when the VI problem is strongly monotone locally. In contrast, the explicit approach works as long as the convergence of the projection method is guaranteed, which only requires monotoncity. (4) We give real-world examples to demonstrate the potential applications of the proposed framework.

**Related Work.** Interventions in games can be modeled as mathematical programs with equilibrium constraints (MPEC), a class of optimization problems constrained by equilibrium conditions, often represented as a VI problem [35]. As our framework casts the equilibria of games as individual layers, it may be viewed as a special class of MPEC. Solving MPEC typically requires calculating the derivatives of the equilibria [19], which is a significant challenge. Another difficulty has to do with the lack of unique equilibrium, a requirement for differentiability [49]. Besides the MPEC approach, recent work also casts the intervention in games as a bi-level reinforcement learning (RL) problem, in which the agents' decision is modeled as a Markov decision process [67].

Learning payoff parameters in games usually relies on the special structures of games [62, 9]. Ling et al. [33] studied how to learn a normal form game using a differentiable game solver layer in an end-to-end framework. The convergence and sensitivity analysis in network games [48] is another example that enable treating game solvers as individual layers. As VI provides a unified formulation for various equilibrium problems arising from games, our work can be regarded as a generalization of these works.

To inject an appropriate inductive bias into the modeling procedure, recent work shows the possibility of embedding differentiable optimization problems as layers in an end-to-end framework [1]. To differentiate through an optimization problem, one could either unroll the numerical computation [17, 4] or implicitly differentiate the optimality conditions, such as the KKT conditions in quadratic programs (QP) [3], the Pontryagin minimum principle in optimal control problems [25], and the Euler-Lagrange equations in least-action problems [36]. As the solution to a VI problem can usually be characterized as a fixed-point equation via a projection operator, which is equivalent to a QP problem, our work is built on some results in [3, 1].

**Organization.**   In § 2, we present a unified optimization framework for learning and interventions in games. § 3 focuses on the VI layer. We first introduce the methods for forward propagation, including projection methods and a Newton-type method. Then, backward propagation methods based on explicit and implicit differentiation are proposed, and their analytical properties are discussed. In § 4, we provide numerical results, inspired by a few real-world applications, that highlight the capabilities of the proposed framework.

**Notation.**   Given a set of scalars $a_i$ or functions $f_i(\cdot)$ with $i$ from a certain indicator set, we denote its vector form as $a = (a_i)^\mathsf{T}$ and $f(\cdot) = (f_i(\cdot))^\mathsf{T}$. We define the Euclidean norm of a vector $a \in \mathbb{R}^n$ as $\|a\|$ and the operator norm of a matrix $A \in \mathbb{R}^{n \times n}$ induced by Euclidean norm as $\|\|A\|\|$. The inner product of two vectors $a, b \in \mathbb{R}^n$ is defined as $\langle a, b \rangle = a^\mathsf{T} b$. The Jacobian matrix of $y \in \mathbb{R}^m$ with respect to $x \in \mathbb{R}^n$ is denoted as $\partial y / \partial x \in \mathbb{R}^{m \times n}$.

## 2   End-to-end design and learning in games

### 2.1   Design and learning in games

We model a system comprised of self-interested agents using game theory. Consider a game played by a set of agents $\mathcal{N}$, where each agent $i \in \mathcal{N}$ selects an *action* $a_i \in \mathcal{A}_i$ and each player's payoff is determined by a function $u_i : \mathcal{A} \to \mathbb{R}$, where $\mathcal{A} = \prod_{i \in \mathcal{N}} \mathcal{A}_i$. The outcome of the game is usually predicted by its Nash equilibrium, where the agents have no incentive to unilaterally deviate from their current *strategies*.

In this paper, we formulate the equilibria of games as parametric VI problems defined below.

**Definition 1** (Parametric VI problem). Given a set $\mathcal{Z}_\lambda \subseteq \mathbb{R}^n$ and a function $F_\lambda : \mathcal{Z}_\lambda \to \mathbb{R}^n$ parameterized by $\lambda \in \mathcal{L} \subseteq \mathbb{R}^m$, a parametric VI problem $\mathrm{VI}(F_\lambda, \mathcal{Z}_\lambda)$ is to find $z^* \in \mathcal{Z}_\lambda$ such that

$$\langle F_\lambda(z^*),\, z - z^* \rangle \geq 0, \quad \text{for all } z \in \mathcal{Z}_\lambda. \tag{1}$$

The equilibrium strategy $z^*$ of various games can be reformulated as the solution to a certain $\mathrm{VI}(F_\lambda, \mathcal{Z}_\lambda)$, where $\mathcal{Z}_\lambda$ is the strategy set and $F_\lambda$ is derived from the payoff functions according to the underlying game structure. For example, when the agent profile $\mathcal{N}$ is finite and the action set $\mathcal{A}_i$ is continuous for all $i \in \mathcal{N}$, methods for establishing the equivalence between Nash Equilibrium and VI, as well as the necessary and sufficient conditions, can be found in [47, 53]. If the action set $\mathcal{A}_i$ is finite, the mixed-strategy Nash equilibrium can be formulated as a VI problem in a similar manner [42]. When the game is placed in a population context, i.e. the agents in $\mathcal{N}$ are infinitesimal, the equilibrium of game is usually called a Wardrop equilibrium, pioneered by Wardrop's first principle [63] in the context of road traffic. The equivalence of Wardrop equilibrium and VI was first established by Dafermos [14]. Below we first give an example of the VI formulation of routing game.

**Example 1** (Routing game). Consider a set of agents traveling from source nodes to sink nodes in a graph with nodes $\mathcal{V}$ and edges $\mathcal{E}$. Each agent aims to choose a route to minimize the total cost incurred. Suppose that $x_e$ is the number of agents choosing edge $e$. The generalized cost on each edge can be set as

$$c_{e,s_e,m_e,\beta_e,\gamma}(x_e) = t_{e,s_e}(x_e) + \gamma m_e + \beta_{e,\theta_e}(x_e), \tag{2}$$

where $t_e$ is the travel time, modeled as a function of $x_e$ and the road capacity $s_e$; $\gamma$ is the time value of money; $m_e$ is a monetary cost; and $\beta_e$, parameterized by $\theta_e$, represents the "hidden" cost that is difficult to measure (e.g., comfort, safety). Denote $\mathcal{X}$ as the set of feasible edge flows satisfying the flow conservation conditions, then the Wardrop equilibrium is equivalent to a VI problem [14]: find $x^* \in \mathcal{X}$ such that

$$\langle c_{s,m,\theta,\gamma}(x^*),\, x - x^* \rangle \geq 0, \quad \text{for all } x \in \mathcal{X}. \tag{3}$$

From the above discussion we can see that the equilibrium of a game depends on the payoff functions. Therefore, to induce a target outcome (e.g., one that maximizes "social welfare") in a game-theoretic system, a central designer can change the payoff functions either by redesigning system parameters or directly through incentives. Before that, the central deisgn must first learn how the agents evaluate the payoff, especially its "hidden" components. To this end, we provide a unified framework for learning and intervention in games as shown in Figure 1, where the equilibrium layer is cast as a VI problem.

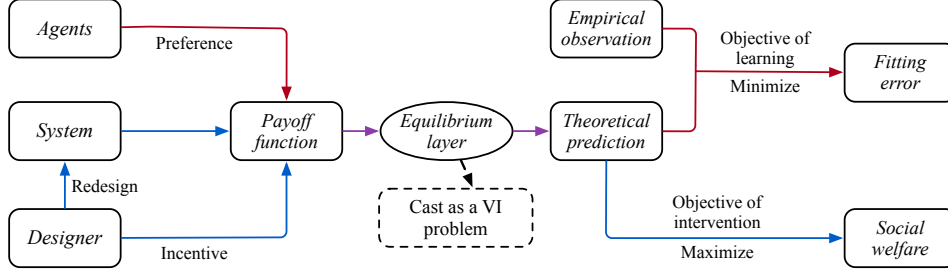

Figure 1: A unified framework for learning and intervention in games.

- The *learning mode* is a regression problem: the designer determines the unknown parameters in agents' payoff functions by minimizing the fitting loss between empirical observations and theoretical predictions.
- The *intervention mode* is a central design problem: the designer modifies the payoffs through incentives or system redesign to induce a target equilibrium.

**Note.**    In many cases, the game designer is expected to design interventions based on learned payoff functions. However, the learning and intervention functions can also be carried out independently. Sometimes, the primary interest is to understand agent behaviors, and hence only the learning mode is needed. Alternatively, when all game inputs are known, the focus would be on intervention.

## 2.2   End-to-end optimization

**Problem formulation.**    Both the learning and intervention modes can be formulated as an end-to-end optimization problem given as follows.

$$\min_{\lambda} \ \ \bar{R}^* = R(\bar{p}^*)$$
$$\text{s.t. } \bar{p}^* = p(\bar{z}, \lambda), \ \bar{z}^* \text{ solves VI}(F_\lambda, \mathcal{Z}_\lambda), \ \lambda \in \mathcal{L}. \tag{4}$$

In both modes, $\bar{z}$ represents the equilibrium strategy, while $F_\lambda$ and $\mathcal{Z}_\lambda$ characterize the setup of game as described in § 2.1. Other variables may have different meanings in the two modes.

- In the *learning mode*, $\lambda$ represents unobservable parameters in payoff functions. Variable $\bar{p}$ is the predicted equilibrium system state that can be observed empirically, and the objective $\bar{R}$ is the fitting loss between $\bar{p}$ and its observation $\hat{p}$. The constraint $\mathcal{L}$ on $\lambda$ may be derived from its physical meaning (e.g. the value of time must be non-negative). If the payoff functions are characterized by more complicated constructs (e.g. a deep network), the constraint may be unnecessary.
- In the *intervention mode*, $\lambda$ represents design parameters. Variable $\bar{p}$ contains equilibrium system states that dictate social welfare, i.e. $-\bar{R}$. The constraint $\mathcal{L}$ on $\lambda$ may be related to the financial and/or physical resources available to the central designer.

When the two modes are integrated in an application, the learning mode learns $\lambda$ in the layer VI$(F_\lambda, \mathcal{Z}_\lambda)$ and shares the learned value with the intervention mode. The intervention mode then directly/indirectly changes $F_\lambda$ to induce a target equilibrium. Below we briefly exemplify the learning and intervention problems in a routing game.

**Example 1** (continued)**.**    Of the three components in the cost function (2), the monetary cost $m_e$ is easy to estimate, and the travel time $t_e$ can be determined from empirical observations, see e.g. the Bureau of Public Roads (BPR) function [11]. The hidden cost function $\beta_e$, as well as the value of time $\gamma$, however, is usually unknown. If the central designer wants to reduce excessive congestion at

equilibrium, she can either adjust $s_e$ or change $m_e$ (e.g., by imposing a toll) for a subset of edges. However, she must first learn $\theta_e$ and $\gamma$ in order to correctly anticipate the overall impact of her action on the generalized cost.

- In the learning mode, decision variables include $\gamma$ and $\theta_e$. Denote the equilibrium flow on edge edge $e$ predicted by (3) as $\bar{x}_e^*$. Suppose that $\hat{E}$ is the set of edges where the flow can be observed. Then the system states used for fitting are the predicted flow $\bar{x}_e$ and the corresponding observed flow $\hat{x}_e$ for $e \in \hat{E}$. The objective can be the squared loss $\bar{R} = \sum_{e \in \hat{E}} (\bar{x}_e^* - \hat{x}_e)^2$.

- In the intervention mode, the decision variables are $s_e$ and/or $m_e$. The social welfare can be measured by the total travel delay, which is determined by the predicted flow $\bar{x}_e^*$ and travel time $t_{e,s_e}(\bar{x}_e^*)$ on each edge $e$. Here, the objective is the total time costs $\bar{R} = \sum_{e \in E} t_{e,s_e}(\bar{x}_e^*) \cdot \bar{x}_e^*$.

**Gradient descent method.** In this paper, we aim to solve (4) using gradient descent methods. At the current point $\lambda^k$, in the *forward propagation*, we need to compute $\bar{R}^k = L(p_\lambda^k)$, where $p_\lambda^k = p(\bar{z}^k, \lambda^k)$ and $\bar{z}^k$ is the solution to $\text{VI}(F_{\lambda^k}, \mathcal{Z}_{\lambda^k})$. While in the *backward propagation*, we need to update $\lambda^k$ in the opposite direction of the gradient of the objective function, specifically

$$\frac{\partial \bar{R}^k}{\partial \lambda^k} = \nabla R(p_\lambda^k) \cdot \left( \nabla_z p(\bar{z}^k, \lambda^k) \cdot \frac{\partial \bar{z}^k}{\partial \lambda^k} + \nabla_\lambda p(\bar{z}^k, \lambda^k) \right). \tag{5}$$

# 3 Differentiable VI layer

## 3.1 Fixed-point formulation of VI

A VI problem can be equivalently formulated as a fixed-point problem via a projection operator [26].

**Definition 2** (Projection operator). The projection operator $\mathcal{P}_{\mathcal{Z}}$ with respect to the Euclidean norm is defined as

$$\mathcal{P}_{\mathcal{Z}}(y) = \arg\min_{y^* \in \mathcal{Z}} \|y^* - y\|. \tag{6}$$

**Proposition 1** (Fixed-point formulation of VI [26]). The point $z^* \in \mathcal{Z}_\lambda$ is a solution to $\text{VI}(F_\lambda, \mathcal{Z}_\lambda)$ if and only if for any $r > 0$, $z^*$ is a fixed point of the mapping $h_\lambda(z) : \mathcal{Z}_\lambda \to \mathcal{Z}_\lambda$ defined as

$$h_\lambda(z) = \mathcal{P}_{\mathcal{Z}_\lambda}(z - rF(z, \lambda)). \tag{7}$$

Proposition 1 implies that finding a solution to $\text{VI}(F_\lambda, \mathcal{Z}_\lambda)$ is equivalent to finding a fixed point of $h_\lambda(z)$. The projection operator $\mathcal{P}_{\mathcal{Z}_\lambda}(y)$ is denoted as $g_\lambda(y)$ hereafter in this paper and the scalar $r$ is omitted in $g_\lambda(y)$ for simplicity.

Based on the fixed-point formulation, the existence and uniqueness conditions of $\text{VI}(F_\lambda, \mathcal{Z}_\lambda)$ can be established under certain monotone conditions of $F_\lambda$ [22, 38]. See Appendix A for more details. In the sequel, we make the following assumptions, under which $\text{VI}(F_\lambda, \mathcal{Z}_\lambda)$ has at least one solution, as well as some other properties for further analysis.

**Assumption 1.** The function $F_\lambda$ is continuous differentiable and monotone for all $\lambda \in \mathcal{L}$.

**Assumption 2.** The set $\mathcal{Z}_\lambda$ is a bounded polyhedral set for all $\lambda \in \mathcal{L}$.

**Assumption 3.** The mapping $(y, \lambda) \mapsto g_\lambda(y)$ is continuously differentiable.

## 3.2 Forward propagation

### 3.2.1 Projection method

The fixed-point formulation of VI implies that one may iteratively project $z$ to $h_\lambda(z)$ until a fixed point is found. Such an idea leads to a class of algorithms for solving VI problems, known as the projection method. A sufficient condition for convergence is established by Dafermos [15].

**Proposition 2** (Convergence conditions for the projection method [15]). Starting with $z^0 \in \mathcal{Z}_\lambda$, the sequence generated by $z^{k+1} = h_\lambda(z^k)$ converges to $\text{VI}(F_\lambda, \mathcal{Z}_\lambda)$ if

$$\|I - r \cdot \nabla_z F_\lambda(z)\| < 1, \quad \text{for all } z \in \mathcal{Z}_\lambda. \tag{8}$$

If $F_\lambda$ is strictly monotone, the convergence condition (8) is satisfied as long as $r$ is sufficiently small. However, when $F_\lambda$ is monotone but not strictly monotone, such a condition sometimes is not satisfied for all $r > 0$. In this case, provided that $F_\lambda$ is co-coercive with module $c$ on $\mathcal{Z}_\lambda$, the sequence also globally converges to the solution to $\mathrm{VI}(F_\lambda, \mathcal{Z}_\lambda)$ if $r < 2c$. See [39] for more details. In the sequel, we also make the following assumption.

**Assumption 4.** For all $\lambda \in \mathcal{L}$, there exists $c > 0$ such that the function $F_\lambda$ is co-coercive with module $c$ on $\Omega_\lambda$.

Under Assumptions 1 and 4, a sufficiently small $r$ can guarantee convergence for projection methods.

The rate of convergence of the projection method is typically linear [8], while better results can be obtained if we scale $z^k$ at each iteration by a positive definite matrix $G^k$ containing first derivative information of $F_\lambda(z^k)$ (in which case it may be viewed as a variant of the Newton's method [8]). In the latter case, finding a suitable $G^k$ is often a nontrivial task.

### 3.2.2 Newton's method

Finding a solution of $\mathrm{VI}(F_\lambda, \mathcal{Z}_\lambda)$ is equivalent to finding a root of the following equation

$$e_\lambda(z) = z - h_\lambda(z) = 0. \tag{9}$$

Now we are ready to present one of our main contributions, a Newton-type method that solves VI problem by directly locating the solution through root-finding. Firstly, as both $g_\lambda(y)$ and $F_\lambda(z)$ are continuously differentiable per assumption, $e_\lambda(z)$ is also continuously differentiable based on the chain rule. At the point $z^k$, the Newton direction $d^k$ can be found by solving

$$\nabla_z e_\lambda(z^k) \cdot d^k = e_\lambda(z^k). \tag{10}$$

Denote $y^k = z^k - rF_\lambda(z^k)$, then

$$\nabla_z e_\lambda(z^k) = I - \nabla_y g_\lambda(y^k) \cdot \left(I - r \cdot \nabla_z F_\lambda(z^k)\right). \tag{11}$$

To obtain $\nabla_y g_\lambda(y^k)$, we need to differentiate through the projection operator $g_\lambda(y)$. As $\mathcal{Z}_\lambda$ is a polyhedral set, $g_\lambda(y)$ can be converted into a QP problem:

$$
\begin{aligned}
z^* = \arg\min_z \quad & \frac{1}{2} z^T z - y^T z \\
\text{s.t.} \quad & A_\lambda z \le b_\lambda, \quad M_\lambda z = q_\lambda.
\end{aligned}
\tag{12}
$$

This allows us to utilize the following result established by Amos and Kolter [3].

**Proposition 3** (Differentiating through a QP problem [3])**.** In the QP problem (12), denote $\nu$ and $\mu \ge 0$ as the dual variables on the equality and the inequality constraints, respectively. Then the derivatives of the optimal solution $z^*$, $\mu^*$ and $\nu^*$ satisfy the following linear equations

$$
\begin{bmatrix}
I & A_\lambda^\mathsf{T} & M_\lambda^\mathsf{T} \\
\mathrm{diag}(\mu) A_\lambda & \mathrm{diag}(A_\lambda z^* - b_\lambda) & 0 \\
M_\lambda & 0 & 0
\end{bmatrix}
\begin{bmatrix}
\mathrm{d}z^* \\
\mathrm{d}\mu^* \\
\mathrm{d}\nu^*
\end{bmatrix}
=
\begin{bmatrix}
\mathrm{d}y - \mathrm{d}A_\lambda^\mathsf{T}\mu^* - \mathrm{d}M_\lambda^\mathsf{T}\nu^* \\
-\mathrm{diag}(\mu^*)\,\mathrm{d}A_\lambda z^* + \mathrm{diag}(\mu^*)\,\mathrm{d}b_\lambda \\
-\mathrm{d}M_\lambda z^* + \mathrm{d}q_\lambda
\end{bmatrix}.
\tag{13}
$$

Based on Proposition 3, we can obtain the Jacobian of $z^*$ with respect to any parameters. To obtain $\partial z^*/\partial y$, we can substitute $\mathrm{d}y = I$ and set other differential terms in the right-hand side to zero. In this manner, the Newton direction $d^k$ can be derived at each iteration. Subsequently, we set

$$z^{k+1} = z^k - d^k, \tag{14}$$

and move to the next iteration. The following theorem establishes the *local* convergence of this iteration.

**Theorem 1** (Local convergence of Newton's method)**.** Suppose that $\mathrm{VI}(F_\lambda, \mathcal{Z}_\lambda)$ admits a solution $\bar{z}^*$. If $\nabla_z e_\lambda(\bar{z}^*)$ is nonsingular, then there exists a neighborhood $\mathcal{B}(\bar{z}^*)$ of $\bar{z}^*$, such that when starting from $x^0 \in \mathcal{B}(\bar{z}^*)$, the sequence generated by (14) converges to $\bar{z}^*$ superlinearly.

*Proof.* See Appendix B.1 for a detailed proof. □

To enable global convergence, we can first employ the projection method to get into the neighborhood of the solution to $\mathrm{VI}(F_\lambda, \mathcal{Z}_\lambda)$. See Appendix B.2 for a globally convergent algorithm and implementation details.

### 3.3 Backward propagation

#### 3.3.1 Explicit differentiation method

According to Proposition 2, the convergence of the projection method is guaranteed by a sufficiently small $r$. Nevertheless, it is usually unnecessary to predetermine $r$. Instead, we can dynamically adjust $r^k$ at each iteration $k$ such that $r^k$ satisfies the convergence condition (2) when $k$ is sufficiently large. Then the sequence generated by $z^{k+1} = h_\lambda(z^k)$ with $r^k$ converges to the solution to $\text{VI}(F_\lambda, \mathcal{Z}_\lambda)$.

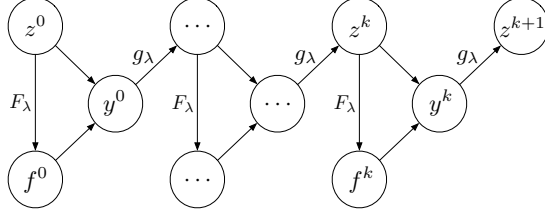

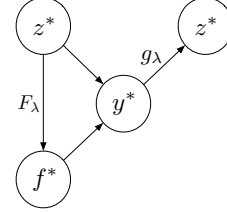

Figure 2: Explicit method.　　　　　　　Figure 3: Implicit method.

The computation graph corresponding to this method is given in Figure 2, where $y^k = z^k - r^k \cdot f^k$ and $f^k = F_\lambda(z^k)$. Based on this graph, by viewing the solver of $\text{VI}(F_\lambda, \mathcal{Z}_\lambda)$ as a collection of an infinite number of projection layers, we are ready to present our first method to differentiate through a VI problem. First, taking the differentials on both sides of $z^{k+1} = h_\lambda(z^k)$ with respect to $\lambda$ gives

$$\frac{\partial z^{k+1}}{\partial \lambda} = \nabla_z h_\lambda(z^k) \cdot \frac{\partial z^k}{\partial \lambda} + \nabla_\lambda h_\lambda(z^k), \tag{15}$$

where

$$\nabla_z h_\lambda(z^k) = \nabla_y g_\lambda(y^k) \cdot \left(I - r^k \cdot \nabla_z F_\lambda(z^k)\right), \tag{16}$$

$$\nabla_\lambda h_\lambda(z^k) = \nabla_\lambda g_\lambda(y^k) - r^k \cdot \nabla_y g_\lambda(y^k) \cdot \nabla_\lambda F_\lambda(z^k). \tag{17}$$

Based on the recursive equation (15), the Jacobian matrix $\partial z^*/\partial \lambda$ can be explicitly derived as the limit of $\partial z^k/\partial \lambda$. At the core of this method is the computation of $\nabla_y g_\lambda(y^k)$ and $\nabla_\lambda g_\lambda(y^k)$, which can still be obtained by Proposition 3. In the backward propagation, however, it is not necessary to explicitly form $\nabla_y h_\lambda(z^k)$ and $\nabla_\lambda h_\lambda(z^k)$. Instead, the Python library `cvxpylayers`[1] can be used to explicitly build the computation graph with projection (QP) layers and enable backward propagation. See [1] for more details on the package.

#### 3.3.2 Implicit differentiation method

If $F_\lambda$ is strictly monotone for all $\lambda \in \mathcal{L}$, then the solution to $\text{VI}(F_\lambda, \mathcal{Z}_\lambda)$ is unique. In this case, the function $z^*(\lambda)$ is directly defined by the fixed-point equation (9), and thus, implicit differentiation can be used to derive $\partial z^*/\partial \lambda$. We first give the sufficient conditions for differentiation.

**Proposition 4** (Conditions for differentiation [57])**.** Suppose that $\text{VI}(F_\lambda, \mathcal{Z}_\lambda)$ admits a solution $\bar{z}^*$ at $\bar{\lambda}$. If $F_\lambda(z)$ is strongly monotone in a neighborhood $\mathcal{B}(\bar{z}^*)$ of $\bar{z}^*$, then in a a neighborhood $\mathcal{B}(\bar{\lambda})$ of $\bar{\lambda}$, for each $\lambda \in \mathcal{B}(\bar{\lambda})$, $z^*$ can uniquely defined from the fixed-point equation $z^* - h_\lambda(z^*) = 0$, and the function $z^*(\lambda)$ defined in this way is differentiable.

Under the differentiation conditions, we are ready to present our second method to perform backward propagation through a VI layer.

**Proposition 5** (Implicit backward propagation)**.** Under the conditions in Proposition 4, we have

$$\frac{\partial z^*}{\partial \lambda} = [I - \nabla_z h_\lambda(z^*)]^{-1} \cdot \nabla_\lambda h_\lambda(z^*). \tag{18}$$

If we want to use (18) to compute $\partial z^*/\partial \lambda$, then we need to differentiate through the mapping $h_\lambda(z^*)$. The computation graph of this method is shown in Figure 3, where $y^* = z^* - r \cdot f^*$ and $f^* = F_\lambda(z^*)$. As $z^*$ is already the solution to $\text{VI}(F_\lambda, \mathcal{Z}_\lambda)$, $r$ can be any positive value. The Jacobian matrix $\nabla_z h_\lambda(z^*)$ and $\nabla_\lambda h_\lambda(z^*)$ can be obtained using the same method as in the explicit method.

### 3.3.3 Discussion

The following theorem establishes the equivalence of the explicit differentiation method and the implicit differentiation method under the strongly monotone condition of $F_\lambda$.

**Theorem 2** (Equivalence of the two backward propagation methods)**.** Suppose that $\text{VI}(F_\lambda, \mathcal{Z}_\lambda)$ admits a solution $\bar{z}^*$ at $\bar{\lambda}$. Denoting $h_\lambda^{(k)}(\cdot)$ as the $k$th composition of $h_\lambda(\cdot)$, if $r$ satisfying the convergence condition (8) for $\bar{\lambda}$, then for each $\lambda$ in a neighborhood $\mathcal{B}(\bar{\lambda})$ of $\bar{\lambda}$, the function $z^\infty(\lambda) = \lim_{k \to \infty} h_\lambda^{(k)}(z^0)$ is well-defined and is the solution to $\text{VI}(F_\lambda, \mathcal{Z}_\lambda)$. If $F_\lambda(z)$ is strongly monotone in a neighborhood $\mathcal{B}(\bar{z}^*)$ of $\bar{z}^*$, the function $z^\infty(\lambda)$ is differentiable at $\bar{\lambda}$. Moreover, starting from $\partial z^0(\bar{\lambda})/\partial \lambda = 0$, the sequence $\partial z^k(\bar{\lambda})/\partial \lambda$ defined by (15) converges to the Jacobian matrix derived from the implicit differentiation method.

*Proof.* See Appendix C for a detailed proof. □

**Comparison.** As shown in Figures 2 and 3, the computation graph of the implicit method has a more efficient "depth". Therefore, when $F_\lambda$ is strongly monotone, the implicit method is more efficient than the explicit method. However, even though the uniqueness of system-level state (i.e., $\bar{p}$ in the general formulation (4)) can be guaranteed under weak conditions, the agent-level strategy (i.e., $\bar{z}$ in (4)) is not necessarily unique. It is well-known that this situation may arise in population games where $F_\lambda$ is monotone but not strongly monotone (hence the term $I - \nabla_z h_\lambda(z^*)$ in implicit differentiation becomes singular). To deal with this problem, Tobin and Friesz [58] proposed to differentiate by constructing a specific solution that is a nondegenerate extreme point of the solution set. However, getting such a solution is an extra burden in the context of end-to-end optimization, because it is not directly available. The explicit method devised in our paper works in a similar manner by differentiating a specific solution $z^*$ generated by the projection method. Yet, since finding $z^*$ is an integrated part of the solution algorithm, no extra effort is needed.

**Extension.** The polyhedral assumption on $\mathcal{Z}_\lambda$ is sufficient for various equilibrium problems. Nevertheless, the explicit and implicit differentiation methods as well as the Newton's method can still work as long as $\mathcal{Z}_\lambda$ is convex. In this case, the method proposed in [2, 1] can be used for differentiating through the projection, which is equivalent to a convex program.

## 4 Numerical experiments

### 4.1 Braess's paradox

Using the network shown in Figure 4, Braess [10] demonstrates that expanding the capacity of edge 5 would increase the total travel time at the Wardrop's equilibrium. Assume that the travel demand from node 1 to 4 is $q$ and the travel time on each edge is given by the BPR function $t_e(x_e) = T_e \cdot \left(1 + 0.15 \cdot (x_e/s_e)^4\right)$, where $T_e$ is the free-flow travel time and $s_e$ is the edge capacity. Figure 5 shows the gradient of the total travel time $TT = \sum_{e=1}^{5} t_e(\bar{x}_e) \cdot \bar{x}_e$ at the equilibrium traffic assignment $\bar{x}_e$ with respect to $s_5$ under different travel demand $q$, using the implicit, the explicit and the numerical method, respectively. The three methods produce the same results, confirming the recent finding [13] that Braess's paradox exists when travel demand is neither too low (little congestion) nor too high (too much congestion). For details please refer to Appendix D.1.

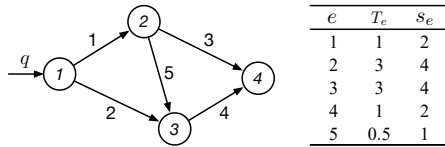

| $e$ | $T_e$ | $s_e$ |
|---|---|---|
| 1 | 1 | 2 |
| 2 | 3 | 4 |
| 3 | 3 | 4 |
| 4 | 1 | 2 |
| 5 | 0.5 | 1 |

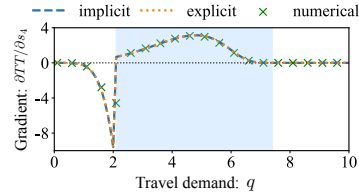

Figure 4: Braess network and its edge parameters.

Figure 5: Gradient of total travel time with respect to the capacity on edge 5.

## 4.2 Transportation system operation in a learning-to-design manner

We then test our methods on a linear city model (Figure 6), where each node represents a business or a residential area. The nodes are linked by roads (driving) and also supported by public transport services (riding). Citizens travel between nodes everyday and we model the choices of citizens using the routing game. To consider the choice of modes and routes simultaneously, we split each node into 4 sub-nodes, namely the starting node "s", the ending node "e", the driving node "v", and the riding node "p". We model the travel costs on driving and riding edges using function (2) in Example 1 and a constant cost on starting edges for public transport, e.g. $1s \rightarrow 1p$, to represent the waiting time.

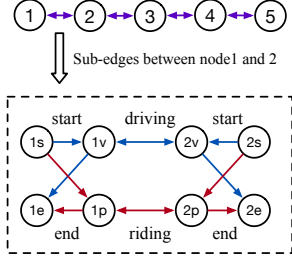

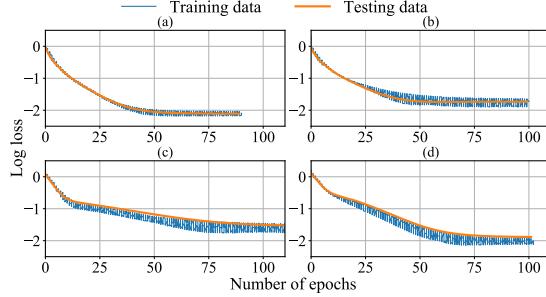

Figure 6: A linear city.    Figure 7: Training process for learning game parameters

**Learning.** We assume that each riding edge $e$ has a hidden cost $\beta_{e,q_e,\tau}(x_e) = \tau \cdot (x_e/q_e)^2$, which indicates the discomfort caused by crowdedness. We first learn $\tau$, $q_e$ and also $\gamma$ (time value of money) based on "observed" traffic flows on the driving and riding edges. We randomly generated $N$ source-sink demand matrices, representing the travel demand in $N$ different periods. We use the true cost functions to generate observations by finding equilibrium traffic flows, and round them to the nearest 0.1. The model is trained using the stochastic gradient decent method. Figure 7 shows the training process under 4 different hyperparameters settings. We report the losses on log scale for better visualization. For details please refer to Appendix D.2.

**Intervention.** We further study how to regulate the transportation system based on the learned cost functions. To encourage transit ridership—widely promoted for sustainability—we consider imposing a congestion toll on driving edges. Specifically, on each riding edge $e$, we design the toll $\pi_e$ to minimize the total travel time, subject to a constraint that bounds the total cost of crowdedness. We solve the toll optimization problem using the steepest descent method with the Armijo-type line search method. First, as a benchmark, when the true cost functions are used, the total travel time is reduced by $10.21\%$ whereas the total crowdeness cost increases by $15\%$ (the preset upper bound). Then, using the cost functions obtained under one of the the four learning modes (a-d in Figure 7), we find the optimal design and test its performance. The travel time savings and the extra cost of crowdedness for the four learning modes are respectively: (a) $11.05\%$, $16.47\%$, (b) $25.79\%$, $54.27\%$, (c) $31.38\%$, $261.48\%$, (d) $8.38\%$, $11.95\%$. For details please refer to Appendix D.2.

## 5   Summary

Over the past decade, artificial intelligence and machine learning have become an increasingly prominent toolbox for understanding social systems [31, 23, 44, 67, 54]. Our work adds into this toolbox a general representation of equilibria of games, and mathematics required to perform forward and backward propagation through it. We hope this work will draw more attention to further developments and applications that could contribute to sustainable development of our shared society.

## Broader Impact

Our work helps understand and resolve social dilemmas resulting from pervasive conflict between self- and collective interest in human societies. The potential applications of the proposed modeling framework range from addressing externality in economic systems to guiding large-scale infrastructure investment. Planners, regulators, policy makers of various human systems could benefit from the decision making tools derived from this work.

## Acknowledgments and Disclosure of Funding

The authors would like to thank the area chair and reviewers for careful reading and constructive suggestions and thank Dr. Liping Zhang, Ruzhang Zhao, Qianni Wang and Lingxiao Wang for providing useful materials and enlightening discussions throughout this project. The research was supported by the US National Science Foundation under the award number CMMI 1922665.

## Footnotes

[1] https://github.com/cvxgrp/cvxpylayers

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
