[Supplementary Material · full.pdf]

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

# A  Existence and uniqueness conditions for VI

We first give the definitions of monotonicity, strictly monotonicity and strongly monotonicity.

**Definition A.1** (Monotone and strictly monotone). A function $F(z)$ is monotone on $\mathcal{Z}$ if

$$\langle F(z_1) - F(z_2), z_1 - z_2 \rangle \geq 0, \quad \text{for all } z_1, z_2 \in \mathcal{Z}, \tag{A.1}$$

and strictly monotone if the inequality above is strict.

**Definition A.2** (Strongly monotone). A function $F(z)$ is strongly monotone on $\mathcal{Z}$ if for some $\alpha > 0$,

$$\langle F(z_1) - F(z_2), z_1 - z_2 \rangle \geq \alpha \|z_1 - z_2\|^2, \quad \text{for all } z_1, z_2 \in \mathcal{Z}. \tag{A.2}$$

We then give the definitions of positive semi-definiteness, positive definiteness and strongly positive definiteness.

**Definition A.3** (Positive semi-definite and definite). A square matrix $A \in \mathbb{R}^{n \times n}$ is positive semi-definite if

$$v^\mathsf{T} A v \geq 0, \quad \text{for all } v \in \mathbb{R}^n, v \neq 0, \tag{A.3}$$

and positive definite if the inequality above is strict.

**Definition A.4** (Strongly positive definite). A square matrix $A \in \mathbb{R}^{n \times n}$ is strongly positive definite if for some $\alpha > 0$,

$$v^\mathsf{T} A v \geq \alpha \|v\|^2, \quad \text{for all } v \in \mathbb{R}^n. \tag{A.4}$$

The following propositions can be used to check whether a matrix is strongly positive definite.

**Proposition A.1.** A square matrix $A$ is strongly positive definite if and only if $A^\mathsf{T} + A$ is positive definite.

The monotonicity of $F(z)$ is closely related to the positive definiteness of $\nabla F(z)$ [46].

**Proposition A.2.** Suppose that $F(z)$ is continuously differentiable on $\mathcal{Z}$ and $\nabla F(z)$ (need not to be symmetric) is positive semi-definite (positive definite), then $F(z)$ is monotone (strictly monotone).

**Proposition A.3.** Suppose that $F(z)$ is continuously differentiable on $\mathcal{Z}$ and $\nabla F(z)$ is strongly positive definite, then $F(z)$ is strongly monotone.

Particularly, if $\nabla F(x)$ is symmetric, then $F(x)$ is strongly monotone if and only if $F(x)$ is strictly monotone.

Eventually, the following propositions provide conditions under which the existence and uniqueness of the solution to VI$(F_\lambda, \mathcal{Z}_\lambda)$ are guaranteed [22, 38].

**Proposition A.4** (Existence condition). If $F_\lambda$ is continuous on $\mathcal{Z}_\lambda$ and $\mathcal{Z}_\lambda$ is compact and convex, then VI$(F_\lambda, \mathcal{Z}_\lambda)$ admits at least one solution.

**Proposition A.5** (Uniqueness condition). If $F_\lambda$ is strictly monotone on $\mathcal{Z}_\lambda$, then VI$(F_\lambda, \mathcal{Z}_\lambda)$ admits a unique solution if one exists. If $F_\lambda$ is strongly monotone, then it always admits one and only one solution.

# B  Details of Newton's method

## B.1  Proof of Theorem 1

*Proof.* As $e_\lambda(z)$ is continuously differentiable and $\nabla_z e_\lambda(\bar{z}^*)$ is nonsingular, $\nabla_z e_\lambda(z)$ is nonsingular in a neighborhood $\mathcal{B}_1(\bar{z}^*)$ of $\bar{z}^*$. Denote $E^k = \|\nabla_z e_\lambda(z^k)\|$, $y^k = z^k - rF_\lambda(z^k)$, and $\bar{y} = \bar{z}^* - rF_\lambda(\bar{z}^*)$. Starting from $z^0 \in \mathcal{B}_1(\bar{z}^*)$, we can recursively get

$$\begin{aligned}
\|z^{k+1} - \bar{z}^*\| &= \|z^k - \nabla_z e_\lambda(z^k)^{-1} \cdot e_\lambda(z^k) - \bar{z}^*\| \\
&\leq E^k \cdot \|\nabla_z e_\lambda(z^k) \cdot (z^k - z^*) - e_\lambda(z^k)\| \\
&= E^k \cdot \|\nabla_z e_\lambda(z^k) \cdot (z^k - \bar{z}^*) - (e_\lambda(z^k) - e_\lambda(\bar{z}^*))\| \\
&= E^k \cdot \|h_\lambda(z^k) - h_\lambda(\bar{z}^*) - (I - \nabla_z e_\lambda(z^k)) \cdot (z^k - \bar{z}^*)\| \\
&= E^k \cdot \|g_\lambda(y^k) - g_\lambda(\bar{y}) - \nabla_y g_\lambda(y^k) \cdot (I - r \cdot \nabla_z F_\lambda(z^k)) \cdot (z^k - \bar{z}^*)\|.
\end{aligned} \tag{B.1}$$

As both $g_\lambda(y)$ and $F_\lambda(z)$ are continuously differentiable, denoting $y = z - rF_\lambda(z)$, there exists another neighborhood $\mathcal{B}_2(\bar{z}^*)$ of $\bar{z}^*$, such that when $z \in \mathcal{B}_2(\bar{z}^*)$ we have

$$
\begin{aligned}
y - \bar{y} &= z - \bar{z}^* - r \cdot (F_\lambda(z) - F_\lambda(\bar{z}^*)) \\
&= (I - r \cdot \nabla F_\lambda(\bar{z}^*)) \cdot (z - \bar{z}^*) + o(\|z - \bar{z}^*\|).
\end{aligned}
\tag{B.2}
$$

Consequently, we have

$$
g_\lambda(y) - g_\lambda(\bar{y}) - \nabla_y g_\lambda(y)(y - \bar{y}) = o(\|y - \bar{y}\|) = o(\|z - \bar{z}^*\|).
\tag{B.3}
$$

Continuing from (B.1), we have

$$
\begin{aligned}
\|z^{k+1} - \bar{z}^*\| &\le o(z^k - \bar{z}^*) + E^k \cdot \|\nabla_y g_\lambda(y^k) \cdot \left(y^k - \bar{y} - (I - r \cdot \nabla_z F_\lambda(z^k)) \cdot (z^k - \bar{z}^*)\right)\| \\
&\le o(\|z^k - \bar{z}^*\|) + r \cdot E^k \cdot \|\nabla_y g_\lambda(y^k)\| \cdot \|F_\lambda(z^k) - F_\lambda(\bar{z}^*) - \nabla_z F_\lambda(z^k) \cdot (z^k - \bar{z}^*)\| \\
&= o(\|z^k - \bar{z}^*\|) + r \cdot E^k \cdot \|\nabla_y g_\lambda(y^k)\| \cdot o(\|z^k - \bar{z}^*\|) = o(\|z^k - \bar{z}^*\|).
\end{aligned}
\tag{B.4}
$$

Therefore, starting from $z^0 \in \mathcal{B}_1(\bar{z}^*) \cap \mathcal{B}_2(\bar{z}^*)$, the sequence converges to $\bar{z}^*$ superlinearly. □

## B.2 Implementation details

To enable global convergence, we first use the projection method to find a point within a sufficiently small neighborhood of the solution to $\text{VI}(F_\lambda, \mathcal{Z}_\lambda)$ and then use the Newton's method to find the solution. Denote $G(z)$ as the gap (or merit) function to $\text{VI}(F_\lambda, \mathcal{Z}_\lambda)$, e.g. [29], a globally convergent method for solving $\text{VI}(F_\lambda, \mathcal{Z}_\lambda)$ is given in Algorithm 1.

---

**Algorithm 1** Projection-Newton method for solving $\text{VI}(F_\lambda, \mathcal{Z}_\lambda)$.

---

**Input:** Initial point $z^0$ and scalar $r^0$, tolerance value $\epsilon_0, \epsilon_1, \delta_0$, and $\delta_1$, control parameters $0 < \alpha_0 < 1$ and $\alpha_1 > 1$
1: Set $k = 0$ and $m^0 = \infty$.
2: **while** $m^k < \epsilon_0$ **do**
3:     Set $z^{k+1} = g_\lambda(z^k - r^k F(z^k))$ and $m^{k+1} = G(z^{k+1})$.
4:     Set $r^{k+1} = \alpha_0 r^k$ if $m^{k+1}/m^k \ge 1 - \delta_1$, and $r^{k+1} = r^k$, otherwise. Set $k = k + 1$.
5: **end while**
6: **while** $m^k < \epsilon_1$ **do**
7:     Compute $H^k = \nabla_z e_\lambda(z^k)$ using (11) with $r^k$. If $H^k$ is singular, find $\eta^k > 0$ such that $H^k + \eta^k I$ is non-singular. Solve $(H^k + \eta^k I) \cdot d^k = e_\lambda(z^k)$ and set $z^{k+1} = z^k - d^k$ and $m^{k+1} = G(z^{k+1})$.
8:     Set $r^{k+1} = \alpha_1 r^k$ if $m^{k+1}/m^k \ge 1 - \delta_2$, and set $r^{k+1} = r^k$, otherwise. Set $k = k + 1$.
9: **end while**

---

Under Assumptions 1 and 4, a sufficiently small $r$ can guarantee convergence for projection methods. Therefore, we dynamically reduce $r$ at each iteration in the projection phase, if the optimality gap does not decrease after the projection. In the Newton's phase, for all positive $r$, the solution to $\text{VI}(F_\lambda, \mathcal{Z}_\lambda)$ satisfies the fixed-point equation (9). Therefore, theoretically, any $r > 0$ can be used to derive the Newton's direction. In practice, we find that a relatively larger $r$ can lead to a faster convergence speed. Therefore, we dynamically increase $r$ when the convergence speed, measured by the decrease of optimality gap after each iteration, is sufficiently small. Meanwhile, if $\nabla_z e_\lambda(z^k)$ is singular, we modify $\nabla_z e_\lambda(z^k)$ by adding a correction matrix $\eta^k I$ to prevent divergence.

## B.3 Additional experiments

We test Algorithm 1 on finding Wardrop's equilibrium. Before the experiments, we first provide some supplementary details of the VI problem given in Example 1. In the graph, denote $\mathcal{W} \subseteq \mathcal{V} \times \mathcal{V}$ as the set of source-sink pairs and $\mathcal{K}$ as the set of paths. Suppose that each $w \in \mathcal{W}$ is associated with $q_w$ infinitesimal agents. Denote $f_k$ as the number of agents choosing path $k \in \mathcal{K}$. Let $M$ be the path-demand incidence matrix and $\Delta$ be the path-edge incidence matrix. Then the feasible region of path flow $f$ and edge flow $x$ are $\mathcal{F} = \{f : f \ge 0, Mf = q\}$ and $\mathcal{X} = \{x : x = \Delta f, f \in \mathcal{F}\}$,

respectively. The Wardrop's equilibrium can be written into the edge-based formulation as in Example 1: find $x^* \in \mathcal{X}$ such that

$$\langle c(x^*),\, x - x^* \rangle \geq 0, \quad \text{for all } x \in \mathcal{X}, \tag{B.5}$$

or equivalently, the path-based formulation: find $f^* \in \mathcal{F}$ such that

$$\langle \Delta^\mathsf{T} c(\Delta f^*),\, f - f^* \rangle \geq 0, \quad \text{for all } f \in \mathcal{F}. \tag{B.6}$$

Here the parameters in $c(x)$ are omitted for simplicity. We test the algorithms on a two-loop city network as shown in Figure B.1. All edges in the network support the driving mode, while the inner loop and the outer loop of the city are also supported by public transport services. We use the same method as in § 4.2 to model the mode and route choices together (splitting each node into 4 sub-nodes). The cost function $c_e(x)$ on each edge has the following form

$$c_e(x) = \begin{cases} T_e \left( 1 + \left( \frac{x_e}{s_e} \right)^2 \right) + \gamma m_e + \tau \left( 1 + \left( \frac{x_e}{q_e} \right)^2 \right), & \text{for driving and riding edges,} \\ w_e, & \text{for starting edges.} \end{cases} \tag{B.7}$$

We set $\gamma = 1$ and $\tau = 1$; the value of parameters are given in Table B.1. The travel demands are generated from independent and identically distributed uniform $U(5, 10)$ distributions. We compare the convergence speed of three methods for finding equilibrium: the gradient-projection (GP) method [24], the projection-Newton (PN) method and the projection method.

- GP method. The GP method is a *specially* designed method that is widely used to find Wardrop's equilibrium. See [24] for more details.
- PN method (Algorithm 1). We implement the PN method on the path-based formulation (B.6) instead of the edge-based formulation (B.5) to improve the efficiency. We set $\epsilon_0 = 10^3$, $\epsilon_1 = 10^{-3}$, $\delta_0 = 10^{-3}$, $\delta_1 = 0.2$, $\alpha_0 = 0.8$, $\alpha_1 = 2$, and start from $r^0 = 0.5$. The Jacobian matrix is derived using the `cvxpylayers` package in Python at each iteration.
- Projection method. We set $\epsilon_0 = 10^{-3}$ and other parameters same as the projection phase in the PN method.

The initial point $f^0$ is derived from the all-or-nothing assignment, i.e. all the agents choose the shortest paths according to the costs $c_e(0)$ for all $e$. For the path set $K$, it is inefficient and also unnecessary to include all the paths at the beginning. Therefore, the path set $K$ is augmented at each iteration to include the current shortest paths. The gap function for $f$ is set as $G(f) = \langle c(\Delta f), \Delta \bar{f} - \Delta f \rangle$, where $\bar{f}$ is derived from the assignment assuming that all of the agents choose the shortest paths based on the edge costs at $f$. We stop the computation after 150 iterations. We test the algorithms under four demand levels (1x, 2x, 3x, and 4x) and the convergence processes are shown in Figure B.2. We see that the PN method is faster than the others. Noted that the GP method is a specialized method for finding Wardrop's equilibrium while the PN method is a general method for solving VIs, this result is a powerful evidence that our Newton's method, directly locating the solution through root-finding, is an efficient method for solving VIs.

Figure B.1: A two-loop city network.

Figure B.2: Convergence process.

| Mode | Edge | $s_e$ | $T_e$ | $m_e$ | $w_e$ | $q_e$ |
|---|---|---|---|---|---|---|
| | v-v (inner loop) | 10 | 0.833 | 0.167 | - | $\infty$ |
| | v-v (outer loop) | 12 | 2.000 | 0.400 | - | $\infty$ |
| Driving | v-v (radial edges) | 15 | 0.700 | 0.140 | - | $\infty$ |
| | s-v | - | - | - | 0 | - |
| | p-p (inner loop) | $\infty$ | 0.917 | 0.023 | - | 20 |
| | p-p (ourer loop) | $\infty$ | 2.200 | 0.055 | - | 25 |
| Riding | s-p (from inner loop) | - | - | - | 1 | - |
| | s-p (from outer loop) | - | - | - | 3 | - |

Table B.1: Edge parameters of the two-loop city network.

# C  Proof of Theorem 2

*Proof.* For each $\lambda$, the function defined as the limiting point of the projection method is

$$z^\infty(\lambda) = \lim_{k\to\infty} z^k(\lambda) = \lim_{k\to\infty} h_\lambda^{(k)}(z^0). \tag{C.1}$$

As $F_\lambda(z)$ is continuously differentiable and $r$ satisfies the convergence condition (8) for $\bar\lambda$, there exists a neighborhood $\mathcal{B}_1(\bar\lambda)$ of $\bar\lambda$ such that

$$\|\|I - r \cdot \nabla_z F_\lambda(z)\|\| < 1, \quad \text{for all } z \in \mathcal{Z}_\lambda, \text{ and } \lambda \in \mathcal{B}_1(\bar\lambda). \tag{C.2}$$

According to Proposition 2, for all $\lambda \in \mathcal{B}(\bar\lambda)$, the sequence $z^k(\lambda)$ converges to a point $z^\infty(\lambda)$ which is the solution to VI$(F_\lambda, \mathcal{Z}_\lambda)$. Therefore, the function (C.1) is well defined. If $F_\lambda(z)$ is strongly monotone, then there exists another neighborhood $\mathcal{B}_2(\bar\lambda)$ of $\bar\lambda$ such that for all $\lambda \in \mathcal{B}_2(\bar\lambda)$, the solution to VI$(F_\lambda, \mathcal{Z}_\lambda)$ is unique. Then based on Proposition 4, $z^\infty(\lambda)$ is differentiable in $\mathcal{B}_1(\bar\lambda) \cap \mathcal{B}_2(\bar\lambda)$. For all $z \in \mathcal{Z}_\lambda$ and $\lambda = \bar\lambda$, the operator norm of $\nabla_z h_\lambda(z)$ satisfies

$$\|\|\nabla_z h_\lambda(z)\|\| \le \|\|\nabla_y g_\lambda(y)\|\| \cdot \|\|I - r \cdot \nabla_z F_\lambda(z)\|\| \le \|\|I - r \cdot \nabla_z F_\lambda(z)\|\| < 1, \tag{C.3}$$

where the second inequality holds because $g_\lambda$ is an Euclidean projection and hence the singular value of $\nabla_y g_\lambda(y)$ can only be 1 or 0. The convergence of $\partial \bar z^k(\bar\lambda)/\partial\lambda$ is then guaranteed by the contraction mapping theorem. As we assume that both $F_\lambda(z)$ and $g_\lambda(y)$ are continuously differentiable, so is $h_\lambda(z)$. As a result, both $\nabla_z h_\lambda(z)$ and $\nabla_\lambda h_\lambda(z)$ are continuous. Hence in the recursive equation (15), set $\lambda = \bar\lambda$ and let $k \to \infty$, we get

$$\frac{\partial z^\infty(\bar\lambda)}{\partial\lambda} = \nabla_z h_\lambda(z^\infty(\bar\lambda)) \cdot \frac{\partial z^\infty(\bar\lambda)}{\partial\lambda} + \nabla_\lambda h_\lambda(z^\infty(\bar\lambda)). \tag{C.4}$$

Therefore, we have

$$\left(I - \nabla_z h_\lambda(z^\infty(\bar\lambda))\right) \cdot \frac{\partial z^\infty(\bar\lambda)}{\partial\lambda} = \nabla_\lambda h_\lambda(z^\infty(\bar\lambda)). \tag{C.5}$$

Eventually, based on Proposition 5, the sequence $\partial \bar z^k(\bar\lambda)/\partial\lambda$ converges to $\partial z^\infty(\bar\lambda)/\partial\lambda$, which equals the Jacobian matrix derived from the implicit differentiation (18).

$\square$

# D  Experimental settings

## D.1  Braess paradox

In this experiment, we test the accuracy of differentiating through a VI problem by quantifying the Braess paradox. We test both the explicit method and the implicit method, and compare the results with the numerical differentiation.

- Explicit method. We implement the projection method (equivalently, the projection phase of Algorithm 1) on the path-based formulation (B.6) to find equilibrium. In the forward propagation, we set $\epsilon_0 = 10^{-4}$, $\delta_0 = 10^{-3}$, $\alpha_0 = 0.8$, and start from $r^0 = 0.5$. We directly build the computation graph using the `cvxpylayers` package.

- Implicit method. We use the same equilibrium solution as in the explicit method. Different from using the path-based formulation (B.6) in the forward propagation, we implement implicit differentiation on the edge-based formulation (B.5). We also use the `cvxpylayers` package to compute the Jacobian matrices.

- Numerical differentiation. We disturb the capacity by $+5\%$ and use finite-difference method to compute the gradient.

## D.2 Transportation system operation

In this experiment, we test the performance of our framework on the operation of a transportation system. The cost function has the same form as (B.7), and the parameters are given in Table D.1. Meanwhile, we set $\gamma = 1$ and $\tau = 1$.

| Mode | Edge | $s_e$ | $T_e$ | $m_e$ | $w_e$ | $q_e$ |
|---|---|---|---|---|---|---|
| Driving | v-v | 10 | 1.0 | 0.25 | - | $\infty$ |
| | s-v | - | - | - | 0 | - |
| Riding | p-p (left to right) | $\infty$ | 1.1 | 0.05 | - | 18 |
| | p-p (right to left) | $\infty$ | 1.1 | 0.05 | - | 22 |
| | s-p | - | - | - | 1 | - |

Table D.1: Edge parameters of the linear city network.

**Learning.** The number of periods is set as $N = 8$. In each period, the travel demands pair are generated from independent and identically distributed uniform $U(5, 10)$ distributions. We use the first 6 periods for training and the last 2 periods for testing. We use the objective function given in the learning mode of Example 1, assuming that the number of agents (flow) on the driving and riding edges can be observed.

In the forward propagation, we use Algorithm 1 to find equilibrium. We set $\epsilon_0 = 1$ and $\epsilon_1 = 10^{-3}$; other parameters are the same as in Appendix B.3. In the backward propagation, we use the implicit method on the edge-based formulation (B.5). We consider two types of parameters initialization strategies: ① high riding costs; and ② low riding costs. For ①, we set $\gamma^0 = 0.2$, $\tau^0 = 1.5$, and $q_e^0 = 10$ for all riding edges. For ②, we set $\gamma^0 = 1.5$, $\tau^0 = 0.2$, and $q_e^0 = 30$. The learning rate for $\gamma$ and $q_e$ are set as $10^{-4}$, and we consider two learning rates for $\gamma$: ⓘ $10^{-3}$, and ⓙ $10^{-4}$. The 4 hyperparameters settings in Figure 7 are set as (a): ① + ⓘ; (b) ① + ⓙ; (c) ② + ⓘ; (d) ② + ⓙ. The learned value of parameters are given in Table D.2.

| | $\gamma$ | $\tau$ | $q_e$ | | | | | | | |
|---|---|---|---|---|---|---|---|---|---|---|
| | | | 1-2 | 2-1 | 2-3 | 3-2 | 3-4 | 4-3 | 4-5 | 5-4 |
| (a) | 0.914 | 1.012 | 17.761 | 21.928 | 18.396 | 22.421 | 17.971 | 22.010 | 18.692 | 22.825 |
| (b) | 0.293 | 0.957 | 18.611 | 23.379 | 18.495 | 22.624 | 17.906 | 22.027 | 19.695 | 24.364 |
| (c) | 0.014 | 1.164 | 23.097 | 28.021 | 20.201 | 25.167 | 20.402 | 24.831 | 21.447 | 27.551 |
| (d) | 1.250 | 1.289 | 20.828 | 24.863 | 19.798 | 24.397 | 20.577 | 24.869 | 19.193 | 23.681 |

Table D.2: Learned value of parameters.

According to Figure 7, (a) produces the smallest fitting loss. Based on Table D.2, the learned values in (a) are also the closest to the true values.

**Intervention.** In the forward and backward propagation, we use the same methods as in the learning mode. Based on the learned cost functions in (a), the intervention mode produces similar results compared with the benchmark. The learned cost functions can also be used for other interventions, ranging from network expansions to public transport service pricing and headway design.