[Reviews · NeurIPS 2020]

Review 1

Summary and Contributions: This paper studies end-to-end learning and intervention in games in order to incentivize agents to change their behaviors to socially desirable outcomes. Their main observation is that games that can be solved through gradient descent The authors do this through variational inequalities and propose two approaches based on explicit and implicit differentiation. In greater detail, they note that the equilibria of games can be cast as variational inequality problems and it is possible to treat the design of games as an end-to-end process where solving the game is only one layer. Additionally, they prove the equivalence of both the explicit and implicit differentiation approaches and present experiments demonstrating the efficacy of their approach. Overall, I enjoyed reading this paper and found its end-to-end approach interesting. Treating mechanism design as a very general VI problem is a natural approach, and splitting their approach into "learning" and "intervention" phases where the mechanism tries to learn agent behavior in a game and then attempts to intervene in order to effect more societally-optimal behavior is a nice idea. Furthermore, their results seem to be nontrivial, although perhaps this is because I am not incredibly familiar with VI as an approach. However, this paper seems more explorational than technically substantial to me in the sense that they established that mechanism design fits into the VI paradigm, but mostly just establish baseline proof-of-concept results (which is not a bad thing by any means; it's just perhaps not as groundbreaking as I expected).

Strengths: See summary.

Weaknesses: See summary.

Correctness: As far as I can tell, yes, but I did not read the appendix.

Clarity: Yes, it is very nicely written.

Relation to Prior Work: Yes.

Reproducibility: Yes

Additional Feedback: 26: aka. --> a.k.a.\ 32: goods --> good Throughout the paper: pointers to the appendix are broken (perhaps point to the full version of the paper or supplementary material explicitly) I have read the author rebuttal and my review is not changed.


Review 2

Summary and Contributions: This paper uses a Variational Inequality (VI) formulation of games to learn fixed but unknown parameters in the games or intervention parameters. The authors propose using the known fixed point of a projection formulation for the VI to enable taking gradients through the VI function (when it is a function). The authors test the techniques on two small congestion games.

Strengths: The use of VI and incorporating that in a differentiable layer is the main strength on this paper. Other than these please look at the weaknesses that I point out.

Weaknesses: The paper has a decent idea but overall the paper is has weak experiments and seems incremental. Below I point out the weaknesses IMO: 1) The experiments are on such small problems - couldn't these small sizes be solved easily by a some generic non-linear solver (like one in Matlab?). Also, there are claims about general game theory but all experiments are on small non-atomic congestion games, where Wardrop's equilibrium properties already brings about so much required structure. What about more complex and large games? 2) There is no new technique as such in the paper. VI and the fixed point formulation are very well-known methods. So is implicit differentiation. Yes, they are used in a nice way (which is the strength of the paper), but I do not see a non-incremental contribution here. 3) The authors call the framework end to end. In previous work, end to end has been used to describe a system that tied together all components from data to end goal (and then maybe differentiates through all of them). However, here the learning and interventions are separate - if they were tied together and differentiated through end to end then the work would be truly end to end.

Correctness: The claims are correct.

Clarity: The paper can be written better. Below I point a few things: 1) When propositions are used to state a well-known result then a reference must be provided (after proposition number) so make it clear that this is from another paper (or book). 2) Example 2 is really a continuing example and not a different example. 3) In the text before and after Equation 2, the subscripts are a but in Eq 2 they are e. 4) In Eq C.3 in Appendix the limit is pulled inside the derivative. Please write the conditions when this is doable and reference some book/paper that states the conditions.

Relation to Prior Work: Yes, the related work is sufficient.

Reproducibility: Yes

Additional Feedback: -----------Post Rebuttal---------- The idea in the paper is good but I would request the authors to reduce the overclaiming as pointed out by reviwers.


Review 3

Summary and Contributions: The paper focuses on a multi-agent problem where multiple agents with unknown payoffs interact with each other strategically or there exists a game designer who can control some of the game parameters. In the former case, we may want to learn the agents' payoff and in the latter case, the game designer wants to design interventions, i.e., change game parameters to induce an equilibrium with desirable outcomes, e.g., maximizing social welfare. The paper shows that these two problems can be solved with a unified framework with variational inequalities. In addition to this unified formulation, the paper proposes an end-to-end optimization framework to solve the formulation, based on a projection operator.

Strengths: The paper is relevant to NeurIPS community and the problem studied is important. The proposed unified formulation is novel to my knowledge and is very interesting. The proposed end-to-end learning framework is also new and it is analyzed in detail.

Weaknesses: Part of the writing can be improved. See details below.

Correctness: There are some over-generalized claims which are not very rigorous. For example, in the introduction, the paper seems to take it for granted that there is a central designer in every game-theoretic system. This is not the case. Clearly in auction-like mechanism design problems there is a explicit system designer, the auctioneer, but this may not be the case for many other problems. Even for the congestion game, one can consider a game just among the $n$ travelers, with the road network given, or consider a game with $n+1$ players, where the first player is the system designer, who is a leader in the game and chooses a strategy first, while the other $n$ players are the followers. You may say that in any game-theoretic system where there are certain game parameters that may be changed by an external agent, we can treat that external agent as a game designer and the intervention problem that the paper considers is the problem faced by this game designer. Also, in Sec 2.2, it is stated that most equilibrium problems can be formulated using VI, but in fact a Nash equilibrium (NE) problem is equivalent to a VI problem if the payoff functions and strategy sets satisfy certain conditions. It would be better to name a few example problems whose corresponding NE problem can be formulated using VI. There are some non-critical errors: Equation 1, it should be >= instead of <=. Typo: Line 156, it should be L instead of T.

Clarity: The writing and organization of the paper can be improved. It is unclear even at the end of Sec 2.1 how exactly is the learning and intervention problem formulated mathematically except for a specific example. In fact, that example is also confusing. In Example 1, the subscripts should be e instead of a in all the notations. Also it is unclear whether and how constrained the intervention space is. Due to the lack for formal formulation, it is unclear from Sec 2.1 whether the final goal of the game designer is to minimize fitting loss or to maximize social welfare, or both. In Sec 2.2., from (4), it seems that in fact the system designer can choose to either learn or to intervene, and that is why there is the learning mode and the intervention mode in Sec 2.1. But Example 1 makes me feel like the designer's final goal is to maximize social welfare and the learning of parameters is just the way to achieve that. It would be better if the meaning of the two modes can be explained more explicitly: one can choose to either learn or to intervene, and your work shows that the learning problem and intervention planning problem can be formulated and solved in the same way. Other minor issues: Line 165 and 174, map->mapping? Line 190, notrivial->nontrivial?

Relation to Prior Work: The paper provides a reasonable coverage of related work and discussion of the differences. After rebuttal: Thank you for providing the clarifications. I would expect more rigorous descriptions in the future version of the paper.

Reproducibility: Yes

Additional Feedback: See above


Review 4

Summary and Contributions: This work looks at changing the parameters of a game such that the equilibrium behavior of players optimizes some given function. The authors propose a solution with two modes. Assuming some functional form for player utilities, the parameters governing the player utilities are fit to match the empirically observed equilibrium behavior. Next, intervention parameters are then adjusted to shift the equilibrium towards a more desirable state. Both modes hinge on relating the equilibrium behavior of the game to the utility and intervention parameters. This is accomplished using the framework of variational inequalities. The solution to a specific class of variational inequalities can be represented either as the explicit limit to an incremental update or as the solution to an implicit equation. The authors derive gradients for both approaches (d[equilibrium]/d[parameter]) allowing them to ultimately optimize the equilibrium state of the game using gradient methods. Experiments on two model traffic networks are examined: one showing the accuracy of the gradient computation and another demonstrating that the proposed approach can be used to optimize a desired criterion, efficiency, using the derived equilibrium gradient.

Strengths: This paper tackles an interesting problem using a principled and, to my knowledge, novel approach. I think this is relevant to the multi-agent community and is nicely tied to real world domains.

Weaknesses: The class of variational inequalities that the proposed solution applies to (strictly monotone) is somewhat restrictive, but given the novelty of the problem and approach, I don't see this as a major limitation. The paper could benefit from a bit more clarity and rigor with respect to variational inequality theory. Post Rebuttal: I urge the authors to consider limiting the scope of their work to strictly monotone variational inequalities. The projection method provably converges in this setting and would obviate the need for discussing many subtleties regarding the monotone setting. Neither the explicit or implicit methods proposed can robustly handle the monotone setting so considering it in the paper is confusing. Drawing the line at strict monotonicity makes it clear where future work must be done. I still believe this is a good submission and should be accepted so my score remains the same.

Correctness: I don't see any problems with the empirical methodology. There are a few claims that seem somewhat misleading, possibly due to lack of clarity in writing. For example, in subsection 3.2.1, after discussing requirements for the convergence of the projection method, the authors state "Therefore, a sufficiently small r can guarantee convergence for projection methods in general". It's unclear what context this should be read in, but this contradicts the monotone setting mentioned earlier. Maybe change "in general" to "for strictly monotone and co-coercive F". Similarly, in subsection 3.3.3, the authors suggest "differentiating a specific solution z* generated by the projection method through the the computation process" for monotone VIs is similar to "constructing a specific solution which is a nondegenerate extreme point of the solution set to VI". However, the former may not generate a solution to the VI whereas the latter obviously does so the claim is at least misleading. This claim is also made on line 72. The canonical saddle point problem, min_x max_y xy, provides a simple monotone VI example where the projection method fails. The authors may want to consider expanding their results to include the extragradient method (I don't see it being much more work, just some more chain rule) as it will allow them to formally handle the monotone setting. As it stands, the results in this paper are not shown to hold for the monotone setting. Lastly, the authors repeatedly claim that "VI provides a unified formulation for MOST equilibrium problems arising from games" [lines 58-60, 85-86, 139]. This hyperbole is unnecessary and unsupported. An analogy would be to say that convex optimization models most optimization problems, clearly a false statement especially in the "age of deep learning". By equation (3), any fixed point is a solution to a VI. The prevailing equilibrium concept in game theory is Nash and, obviously, not all fixed points are even Nash equilibria.

Clarity: The paper is generally well written. The definition of a Nash equilibrium is mentioned on page 3 but I didn't see it connected to VIs. The connection between monotone VIs and Nash should be made formal with a citation [1,2]. Please point out what "f", the system state, represents in Example 2. I'm confused about how learning is conducted in experiments. Line 294 says "the model was trained". What model is used to learn the parameters of the game (edges)? Are the parameters not just fit directly to the data? Why not just use the mean observed on each edge. This mode of the learning was still not clear after reading D.2. And why is a log loss reported in Figure 7? They are real valued parameters, no? [1] "Projected Dynamical Systems and Variational Inequalities with Applications"; Nagurney & Zhang; '12 [2] "Convex Optimization, Game Theory, and Variational Inequality Theory"; Scutari, Palomar, Facchinei, Pang; '10

Relation to Prior Work: Prior work in game theory and classical approaches to mechanism design and intervention are discussed. The paper could benefit from citing more recent end-to-end work on interventions such as [3]. [3] "The AI Economist: Improving Equality and Productivity with AI-Driven Tax Policies"; https://arxiv.org/abs/2004.13332 After Rebuttal: Thank you for pointing the citation out. I see it in the final summary paragraph now.

Reproducibility: Yes

Additional Feedback: Eqn (1): I believe the inequality is flipped. The utility at Nash should be >= the utility at any other deviation. line 128: x_a refers to edge e? Both a and e subscripts are used in this paragraph. Shouldn't they all be one or the other? line 156: what is "T" in "T(f^k_{\lambda})"? line 190: "an suitable" --> "a suitable" Eqn (18): I believe -r grad F should have parentheses around it line 223: "f^k = F(z^k)" is an abuse of notation. "f" has already been defined in Eqn (4) line 234: "strictly convex" --> "strictly monotone" for map F line 284: "Citizens travels" --> "Citizens travel" line 286: "node'e'" --> "node 'e'"

[Author Response · NeurIPS 2020]

We would like to thank all reviewers for reading our paper and providing constructive comments. Upon acceptance, we will proofread and fix all editorial issues. Below we first address a few common concerns.

**Learning mode and intervention mode.** In most cases, the game designer is expected to first learn about the agents (e.g. payoff functions) and then use the information (shared weights in the equilibrium layer) to design interventions. Sometimes, the primary interest is to understand agent behaviors, and hence only the learning mode is needed. Alternatively, when all game inputs are known, the focus is on the intervention mode. Thus, the two modes are both independent and integrated from the application point of view. In the final version, we will (i) explain in § 2.1 how these two modes work in different contexts; and (ii) emphasize this point in the mathematical formulation presented in § 2.2.

**Relation between VI and equilibrium.** We agree that it is neither rigorous nor necessary to assert that "most" equilibrium problems can be formulated as VI. Suffice it to say that VI is a powerful tool to study various equilibrium problems. In the final version we will (i) stress the connection between monotone VI problems and equilibrium problems with recommended references; and (ii) provide more examples of the equilibrium problems that can be taclked by VI.

**Apply ML in real-world domains.** Our work is inspired by the current interests on complex optimization-based layers. Although the field has laid a solid theoretical foundation, it has yet to fully exploit the connection between this powerful architecture and real-world problems. Our work not only adds a novel theoretical concept into the rapidly growing toolbox, but also demonstrates the potential of applying the end-to-end framework with VI layers in games.

**Reviewer 1.** Thanks for your generally positive view of our approach. We would like to emphasize the technical contribution of our work as follows. It is the first to treat VIs as individual layers in the end-to-end framework. We propose (i) a new algorithm (tested in Appendix B.3) for solving VI problems in forward propagation and (ii) a new sensitivity analysis method for VI, which views VI problem as a network of infinite number of QP layers, for backward propagation. Both methods are novel for VI problems.

**Reviewer 2.** Thanks for your comments and suggestions. Upon acceptance, we will (i) carefully separate well known results included in the analysis from our own results, (ii) treat the second example as a continuing example, and (iii) clarify the conditions used in (C.3). Below are our responses to the weaknesses identified in your report.

**First**, the main focus of our work is to establish the theoretical foundation of differentiable VI layers and explore how such an end-to-end framework can be used for learning and intervention in games. Thus, the experiments focus on small examples to validate the methodologies. As equilibrium layers are much more complicated than traditional layers, it cannot be solved by generic non-linear solvers in MATLAB. **Second**, we respectively disagree with the opinion that the contribution of our work is weakened by the lack of new mathematical techniques. In fact, we propose, analyze and validate two new algorithms, one for solving VI problems and the other for sensitivity analysis, as an integral part of the new framework. We also respectively disagree that Wardrop's equilibrium brings about too much required structure, because it is just a special case of the VI based layers. The proposed framework is not restricted in any way by the structure that comes with that equilibrium problem. **Third**, as explained above, the learning and intervention modes are independent in the sense of objective functions and only integrated from the application point of view. Both modes are end-to-end, because the equilibria of games are integrated into the framework as individual layers. The learning mode itself can be viewed as a generalization of the end-to-end learning framework proposed in reference [33].

**Reviewer 3.** Thanks for your constructive comments. Some of your concerns are addressed in the general response. Below is the answer to an important question raised in your comments.

**Central designer.** We agree that in some cases a game may not have a "natural" designer like the auctioneer. We define the central designer as an authority whose action can influence the outcome of the game. In congestion game, for instance, the owner of the road network (typically the "government") has the power to levy toll on roads, or implement control measures. These actions can be expected to affect how agents behave and eventually the evolution of the game. We believe that in most (if not all) games, an external designer could be created, if only virtually, to play such a role.

**Reviewer 4.** Thanks for your positive view of our work. We will clarify the description about convergence in § 3.2.1. The paper about AI Economist is actually reference [64], and we will add more details in the literature review.

**The explicit differentiation method and the cited method.** Both methods employs a specific solution in the feasible set (typically a polyhedron) to deal with the non-uniqueness issue. In the cited method, it has to be a nondegenerate extreme point. However, if such a solution is not available from the equilibrium algorithm, it has to be obtained separately, which is a burden in the end-to-end optimization. Since our method unrolls the projection method, it only works when $F_\lambda$ is monotone and the projection method converges. We will clarify this point in the final version. Thanks for your advice on how to expand our results to the general monotone setting.

**Example 2 and the experiment.** The formulation of learning mode in the experiment is given in Example 2. In this case, $f$ represents the number of agents on each edge (flow), which is the only observable state. Typically, it is difficult to directly measure/observe the underlying parameters, such as the time value of money and the coefficients in the hidden costs functions. Instead, they are estimated by minimizing the loss, i.e., the sum of squared errors of the equilibrium flows. We report the losses on log scale for better visualization in Figure 7.

[Meta-Review · NeurIPS 2020]

At the end of the review, rebuttal, and discussion phases, three of the four referees (all of whom are knowledgeable in the field) recommended that the paper be accepted. The other referee (R2) felt the paper was not quite ready for publication at NeurIPS in its current form. From the reviews and the reviewer discussion after the author rebuttal, it appears that the four reviewer see similar strengths and weaknesses of the paper, but assessed the paper differently based on these reviews. I believe that R2 stated this well in the discussion phase: R2: ”After reading other reviews (and rebuttal) I feel that my concerns are actually similar to other reviewers, mainly that the paper has a decent idea but (a) under-developed as the evaluation is very basic and (b) over-claiming at a number of places in the paper. Then, I believe that the scores differ due to differing subjective opinion and IMO I maintain the negative view because 1) This paper needs to demonstrate more in terms of experiments, my belief is that an under-developed idea should get be forced to develop more. 2) Overclaiming will hopefully be fixed by the authors but overclaiming, if it stays in the paper, can be harmful for any future paper in this area." In the discussion phase, the other reviewers confirmed concerns about “overclaiming.” For example, R4 expressed concern that the authors did not effectively establish the boundaries of their claims (and indeed made the claims seem broader than they actually were). In R4’s words: “In the rebuttal, the authors appear to double-down on asserting that their approach makes sense as long as the problem is monotone: `Since our method unrolls the projection method, it only works when Fλ is monotone and the projection method converges’ despite my providing an example where the projection method diverges from the Nash equilibrium (min_{x in [-1,1]} max_{y in [-1,1]} xy), i.e., convergence does not imply Nash or epsilon-Nash. The authors need to back off the monotone assumption; their statements still hold with the strictly-monotone assumption which 1) does not weaken the impact of the paper significantly and 2) makes clear where future work needs to be done.” I do note that the authors sent a message asking that R2’s review be discounted due to a variety of incorrect claims about the paper. I agree with the authors' arguments that R2 does appear to have misunderstood or misjudged some aspects of the paper, but also believe (and the other reviewers confirm) that some of R2’s “criticisms” are justified. All this said, the reviewers did all see value in the paper. Even with the stated weaknesses of the submitted paper, three of the four reviewers still thought the paper should be accepted. I am somewhat on the fence, but edge toward recommending “accept (poster).” I hope that the authors pay special attention to making sure their claims are carefully and properly scoped in the final version of their paper, along with addressing the other comments made by the reviewers.